# Clinical Usefulness of Surgical Resection Including the Complementary Use of Radiofrequency Ablation for Intermediate-Stage Hepatocellular Carcinoma

**DOI:** 10.3390/cancers15010236

**Published:** 2022-12-30

**Authors:** Hideko Ohama, Atsushi Hiraoka, Fujimasa Tada, Kanako Kato, Yoshiko Fukunishi, Emi Yanagihara, Masaya Kato, Hironobu Saneto, Hirofumi Izumoto, Hidetaro Ueki, Takeaki Yoshino, Shogo Kitahata, Tomoe Kawamura, Taira Kuroda, Yoshifumi Suga, Hideki Miyata, Jun Hanaoka, Jota Watanabe, Hiromi Ohtani, Masashi Hirooka, Masanori Abe, Bunzo Matsuura, Tomoyuki Ninomiya, Yoichi Hiasa

**Affiliations:** 1Gastroenterology Center, Ehime Prefectural Central Hospital, Matsuyama 790-0024, Japan; 2Department of Gastroenterology and Metabology, Ehime University Graduate School of Medicine, Toon 791-0295, Japan

**Keywords:** hepatocellular carcinoma, BCLC-B, radiofrequency ablation, surgical resection, collaboration

## Abstract

**Simple Summary:**

Transarterial chemoembolizaion or systemic therapy are recommended for intermediate-stage hepatocellular carcinoma (HCC), but the curative potentials are not high. This study aimed to elucidate the clinical usefulness of surgical resection (SR) including the complementary use of radio frequency ablation (RFA) for intermediate-HCC. Forty-five were treated with SR only and 25 were treated with SR and RFA (Comb). There were no significant differences between them in regard to RFS (median 17.7 months vs. 13.1 months, *p* = 0.36), OS (66.5 months vs. 72.0 months, *p* = 0.57). An acceptable five-year cumulative survival rate was obtained in both groups (54% vs. 64%). This retrospective study found no significant differences for RFS or OS between the present SR and Comb groups with intermediate-HCC.

**Abstract:**

Background/Aim: For intermediate-stage hepatocellular carcinoma (HCC) (Barcelona Clinic Liver Cancer [BCLC]-B) cases, transarterial chemoembolization (TACE) is recognized as the standard treatment, while systemic therapy is recommended for TACE-unsuitable HCC. However, because the curative potential is not high, this study was conducted to elucidate the potential outcomes of surgical resection (SR) for BCLC-B HCC cases. Materials/Methods: From January 2000 to July 2022, 70 patients with BCLC-B HCC treated with surgery as the initial treatment were enrolled (median age 67.5 years, beyond up-to-7 criteria 44). Forty-five were treated with SR only (SR group), while twenty-five underwent that with complemental radiofrequency ablation (RFA) (Comb group). Recurrence-free survival (RFS) and overall survival (OS) were retrospectively evaluated in both groups. Results: The median albumin–bilirubin (ALBI) score was better in the SR as compared with the Comb group (−2.74 vs. −2.52, *p* = 0.02), while there were no significant differences between them for median RFS (17.7 vs. 13.1 months; *p* = 0.70) or median OS (66.6 vs. 72.0 months *p* = 0.54). As for those beyond up-to-7 criteria, there were no significant differences for median RFS (18.2 vs. 13.0 months; *p* = 0.36) or median OS (66.5 vs. 72.0 months; *p* = 0.57). An acceptable five-year cumulative survival rate (>50%) was obtained in both groups (54% vs. 64%). Conclusion: This retrospective study found no significant differences for RFS or OS between the present SR and Comb groups with BCLC-B HCC. When possible to perform, the outcome of SR for BCLC-B is favorable, with a five-year survival rate greater than 50%.

## 1. Introduction

Hepatocellular carcinoma (HCC) is the sixth most commonly diagnosed cancer worldwide and third leading cause of cancer-related death [1]. For staging, the Barcelona Clinic Liver Cancer (BCLC) system is one of the leading treatment algorithm [2], which states that surgical resection (SR) is recommended for BCLC-0 (very early stage) and BCLC-A (early stage), while transarterial chemoembolization (TACE) is recommended for BCLC-B (intermediate stage) and systemic treatment for BCLC-C (advanced stage). The 2022 version of the system further stratifies BCLC-B stage patients into three groups according to tumor burden and liver function [2]. The first of those subgroups includes patients with well-defined HCC, who are recommended for liver transplantation if able to meet extended liver transplantation criteria [3]. The second subgroup includes those who have preserved portal flow, are defined as having a tumor burden, and have feasible selective access to feeding, for whom transarterial chemoembolization (TACE) is recommended [2]. As for the third subgroup within BCLC-B, that includes patients with diffuse, infiltrative, and extensive HCC involvement, with systemic treatment recommended [2].

Over the past decade, there have been significant advances in diagnostic methods, surgery techniques, and perioperative care related to surgical resection of HCC. A recently reported meta-analysis showed that laparoscopic hepatectomy for HCC offers advantages over open resection in terms of in-hospital complication rates, less blood loss, and shorter hospital stays [4]. Furthermore, major hepatectomy procedures are increasingly being performed for large multinodular tumors, as well as tumors with major vasculature invasion [5]. Additionally, some reports have noted acceptable long-term outcomes after resection of HCC, even beyond the current guidelines [6,7,8]. However, when a BCLC-B HCC patient is treated with only SR, there is a risk of insufficient residual liver volume. Additionally, should treatment for BCLC-B HCC with only radiofrequency ablation (RFA) be attempted, there is a risk of failure of radical cure when the HCC diameter is too large. On the other hand, previous studies have reported that the combination of SR and RFA for multifocal HCC may increase the cure rate in surgery cases [9,10], although supporting evidence and consensus regarding the efficacy of SR or that in combination with RFA for treatment of BCLC-B HCC are lacking. The aim of the present study was to evaluate the therapeutic efficacy of SR, including when used in combination with RFA, as initial treatment for BCLC-B HCC. 

## 2. Materials and Methods

The entire study protocol was approved by the Institutional Ethics Committee of Ehime Prefectural Central Hospital. From April 2000 to July 2022, 661 patients with naïve HCC were treated with SR at Ehime Prefectural Central Hospital. After excluding patients with performance status 1 or worse, with a single HCC nodule or ≤3 nodules each ≤3 cm, and/or with extrahepatic spread, 70 with BCLC-B HCC were enrolled in this study (median 68 years old, males 54, beyond up-to-7 criteria 44). The patients were divided according to treatment into those treated with SR only (SR group) (*n* = 45), and those who received a combination of SR and RFA (Comb group) (*n* = 25). Data regarding age, gender, HCC etiology, tumor markers as alpha-fetoprotein (AFP) and des-gamma-carboxy prothrombin (DCP), the largest HCC diameter, the number of HCC, and hepatic function were obtained.

The diagnosis of HCC was based primarily on the findings of dynamic computed tomography (CT) and/or EOB-DTPA-enhanced magnetic resonance imaging (EOB-MRI). Tumor diameter was measured using either early or late phase of the images, while staging of tumor was defined according to the BCLC strategy [2]. Hepatic function was assessed by Child-Pugh classification [11], albumin–bilirubin (ALBI grade) [12,13], and modified ALBI (mALBI) grade [14], for which ALBI grade 2 was divided into two sub-grades (mALBI 2a and 2b) with an ALBI score −2.27 as the cut-off value.

When multifocal hepatic tumors deemed unresectable by traditional standard criteria due to either HCC location or number of tumors, patients were properly informed of the treatment options including the guideline-based treatment, and if they wanted the combination of surgery and RFA, they received that treatment; if they did not, they received TACE or systemic therapy as per the guidelines (Figure 1). SR was performed either an open or laparoscopic approach. If RFA was unsuitable due to the location and size of the tumor but resectable, the tumor was treated with SR. RFA was performed under ultrasonography guidance, but if the target tumors were in easy location for performing RFA (e.g., surface), RFA was performed during resection procedure. RFA was only performed for HCC within 3 cm and within three nodules. Otherwise, RFA was performed before then if the waiting time to SR was long, or as soon as the patient’s condition stabilized after SR if the waiting time to SR was short. The ablation area was designed so that a broad non-staining area appeared across the entire margin, compared with the low-density areas in late phase of CT findings, or as low-intensity areas in the hepatobiliary phase shown on MRI before the treatment. In case of incomplete ablation, the protocol was to add sessions until complete ablation. The endpoints of the study were recurrence-free survival (RFS) and overall survival (OS). RFS was defined as the time from the date of treatment for HCC to the date of diagnosis of recurrence and OS was defined as the time from the date of treatment for HCC to date of death for any reason. Complications associated with treatments were recorded and analyzed.

### Statistical Analysis

All data used were accessed through a database application. Statistically significant differences were analyzed using Student’s *t*-test or a Mann–Whitney U test as appropriate. OS and RFS curves were generated using the Kaplan-Meyer method and compared by a log-rank test. 

The Comb group had potential selection bias, as patients with poor liver function and HCC in both lobes tended to be selected. To prevent uncorrectable bias, inverse probability weighting (IPW) was used to determine by arbitrary criteria that the propensity scores for this study were approximately the same. Probabilities (propensity) for the SR and Comb groups were calculated by logistic regression analysis using a set of covariates that may affect RFS and OS. Clinical factors for SR and Comb patients with a *p* value <0.05 were subjected to multivariate analysis, with a *p* value < 0.05 used to indicate significance for HCC location and ALBI score. IPW was defined as 1/(propensity score) in the Comb group and 1/(1-propensity score) in the SR group. Differences in RFS and OS were evaluated by an IPW-adjusted log-rank test. 

All statistical analysis were performed using EZR [15] ver. 1.53 (Saitama Medical Center, Jichi Medical University, Saitama, Japan), a graphical user interface for R (The R Foundation for statistical Computing, Vienna, Austria), or more precisely, a modified version of R commander designed to add statistical functions frequently used in biostatistics.

## 3. Results

### 3.1. Patient Characteristics

In the SR group, median age was 67 years [interquartile range (IQR): 64–75] and there were 33 males (73.3%), while in the Combi group median age was 68 years (IQR: 62–73) and there were 21 males (84.0%). There was no significant difference regarding etiology between the groups. The median levels of alanine aminotransferase, platelets, alpha-fetoprotein (AFP), and des-gamma-carboxy prothrombin (DCP) in the groups were similar. Median ALBI score in the SR group was significantly better than that in the Comb group [−2.74 (IQR: −3.02 to −2.57) vs. −2.52 (IQR: −2.90 to −2.26); *p* = 0.021], while median FIB-4 index in the SR group was also significantly better [3.28 (IQR: 2.30–3.98) vs. 4.18 (IQR: 3.01–6.92); *p* = 0.031]. There were not significant differences regarding median maximum tumor diameter and number of tumors between the groups, although HCC was significantly more extensive in both lobes in the Comb group (76% vs. 35.6%, *p* = 0.002) (Table 1).

In the SR group, 14 patients had tumors localized to one segment and 15 patients to one lobe. Sixteen patients had tumors in both lobes, two underwent the hemihepatectomy and limited resection, two underwent central bisegmentectomy and limited resection, one underwent central bisegmentectomy, one underwent segmentectomy and limited resection, three underwent segmentectomy, one underwent subsegmentectomy and limited resection, and five underwent limited resection. 

In the Comb group, HCC located in both lobes in 19 patients. All tumors treated with RFA were under 3 cm. There were 21 patients with one lesion, 3 patients with two lesions, and 1 patient with three lesions who underwent RFA. SR was performed for other lesions. Ten patients underwent RFA during resection. Four patients underwent RFA after SR, with a median time between SR and RFA of 70 days (IQR: 53–75). The other patients (*n* = 11) underwent preoperative RFA, with a median time between RFA and SR of 31 days (IQR: 24–38). All RFA was finished in a single session. Local recurrence, defined as the appearance of HCC bordering the ablation area within 2 years, was seen in 2 of 25 patients (8%).

### 3.2. RFS and OS

The median follow-up period after enrollment was 41.0 months (IQR: 14.1–80 months) in the SR group and 40.8 months (IQR: 20.2–78.3 months) in the Comb group. Median RFS was not significantly different at 17.7 (95% CI: 10.3–22.1) and 13.1 (95% CI: 9.0–18.9) months (*p* = 0.70) (Figure 2a). The three-year cumulative non-recurrence rate was 21% in the SR group and 19% in the Comb group, and the five-year cumulative non-recurrence rate was 25% and 19%, respectively. Furthermore, median OS was 66.6 (95% CI: 37.7-NA) and 72.0 (95% CI: 37.5–100.0) months, respectively, also not significantly different (*p* = 0.544) (Figure 2b). In the SR group, the three- and five-year cumulative survival rates were 70% and 54%, respectively, as compared with 77% and 65%, respectively, in the Comb group.

Results of multivariate analysis indicated ALBI score, FIB-4 index, and HCC location as significant factors. Therefore, analysis after adjustment with IPW for those factors was performed, with no significant difference found between the SR and Comb groups regarding median RFS (17.7 vs. 12.7 months, *p* = 0.37) or median OS (66.6 vs. 67.5 months, *p* = 0.65) (Figure 3).

### 3.3. Complications Associated with Treatment

In the SR group, complications leading to prolonged hospitalization were noted in four cases (8.9%), including one each with acute respiratory failure from interstitial pneumonia, massive ascites, acute pneumonia with sepsis, and appetite loss. There were three (12%) cases with complications in the Comb group; cellulitis, massive ascites, and acute cholecystitis. The rate of complications was not significantly different between the groups (*p* = 0.694).

### 3.4. Sub-Analysis: Outcome for Cases beyond the up-to-7 Criteria

Twenty-nine patients in the SR group and fifteen in the Comb group had tumors classified as beyond the up-to-7 criteria. There were no significant differences regarding median age, gender, etiology, alanine aminotransferase level, platelets, AFP, DCP, or numbers of HCC between the groups regarding those cases. The largest HCC diameter was not significantly difference between the SR (6.3 cm, 95% CI: 5.3–8.8) and Comb (5.5 cm, 95% CI: 4.85–6.35) groups, although it was larger than that in the overall group and the number of HCC tumors was greater. Median ALBI scores were similar in the SR (−2.74, 95% CI: −3.07 to −2.57) and Comb (−2.52, 95% CI: −2.95 to −2.27) (*p* = 0.134) groups. In the Comb group, HCC invasion was significantly more extensive in both lobes (86.7%) as compared with the SR group (34.5%) (*p* = 0.004) (Table 2).

For the SR and Comb groups, median RFS was not significantly different at 18.2 (95% CI: 8.5–34.3) and 13.0 (95%: CI 5.9–18.9) months, respectively, (*p* = 0.36) (Figure 4a). The three-year cumulative non-recurrence rate was 28% and 18%, respectively, and the five-year cumulative non-recurrence rate was 22% and 18%, respectively. Median OS was 66.5 (95% CI: 29.4–NA) months in the SR group and 72.0 (95% CI: 30.2–86.2) months in the Comb group, also not significantly different (*p* = 0.57) (Figure 4b). The three-year cumulative survival rate was 72% in the SR group and 75% in the Comb group, while the five-year cumulative survival rate was 54% and 64%, respectively, with survival shown in more than 50% of the patients in each group.

## 4. Discussion

In the present cohort, median RFS and OS in the SR group were 17.7 and 66.6 months, respectively, while those were 13.1 and 72.0 months, respectively, in the Comb group. Analysis after adjustment with IPW showed no significant difference regarding RFS or OS between the groups. From the viewpoint of therapeutic effect, there was no significant difference between the groups in the present study. 

BCLC-B patients who are not indicated for liver transplantation, and have preserved portal flow and feeding arteries are easily selected as candidates for TACE. The BCLC strategy suggests that the expected median survival for such patients is ≥2.5 years [2]. Prince reviewed 101 different studies that included a total of more than 10,000 patients and noted that the five-year overall survival rate for BCLC-B HCC patients after TACE was 32%, with a median survival time of 19.4 months (95%CI: 16.2–22.6) [16]. The present results showed that median OS was 66.6 months in the SR group and 72.0 months in the Comb group, both greater than 2.5 years and more favorable as compared with TACE treatment for BCLC-B.

Kim [17] reported that patients with complete response following the first chemoembolization had the longest OS (70.2 months), followed by those with complete response after two sessions (40.6 months) and then by patients with partial response as the best response (23.0 months) (*p* < 0.01). Both the initial and best response to TACE are important to improve prognosis, while Bolondi’s subclassification [18] and the Kinki criteria [19] suggest that patients classified as beyond up-to-7 criteria are not suitable for TACE. Regarding the Kinki criteria, Arizumi [20] found that the median time to untreatable progression was 25.7 months (95% CI: 19.3–37.3) for patients classified as substage B1 (within up-to-7) and 16.4 months (95% CI: 13.1–20.2) for patients classified as substage B2 (beyond up-to-7) (*p* = 0.005). In other studies of Japanese cohorts, OS for intermediate-stage HCC patients classified as beyond the up-to-7 criteria and treated with TACE was found to range from 20.4–27.6 months [21,22]. In the present beyond up-to-7 criteria cases, median OS was 66.5 months in the SR group and 72.0 months in the Comb group, comparable with the results for all BCLC-B patients noted above.

BCLC-B defined as multifocal HCC with preserved liver function, no cancer related symptoms and no vascular invasion or extrahepatic spread [2]. This definition does not mention tumor localization. Additionally, the BCLC strategy does not consider treatment combinations. However, as in this study, even if there are multiple HCCs, they may be localized to one segment or lobe. This study shows that even when HCC has spread to both lobes, it may be possible to preserve residual liver volume by combining segmentectomy and limited resection, or by combining SR for large lesions and RFA for small lesions.

Recently in the field of hepato-pancreato-biliary surgery, there have been significant advances in diagnostic methods, surgical techniques including laparoscopic surgery, and perioperative care options [23,24]. Some studies have suggested that resection of BLCL-B HCC in selected cases is safe and technically feasible [6,8,25]. In an observational study of patients with HCC, five-year survival following SR for BCLC-B HCC was 57%, which was similar to the five-year survival rate of 61% for patients with BCLC-0/A HCC [25]. Tsilimigras reported that the cure fraction of patients undergoing SR for BCLC-B HCC was 37.6% [26], suggesting that even patients with BCLC-B HCC can demonstrate a complete response. Furthermore, other studies have examined SR in combination with intraoperative RFA for cases with multifocal hepatic tumors [9,10]. In a study of BCLC-B patients, Espinosa reported propensity-score-based analysis findings showing a favorable OS of 60 months (95% CI: 49.1–70.9) for patients that received a combination of SR and RFA, while that was 39.9 months (95% CI: 34.2–45.6) for those who received TACE [27]. Tada also reported five-year cumulative overall survival rates of 68.2% in an SR group, and 59.6% in a combined SR and RFA group of patients with multiple (≤5) HCC tumors, which was not significantly difference (*p* = 0.329) [28]. Similarly, the present study found good results with a five-year survival rate of greater than 50% in both the SR and Comb groups. Although there have been few reports comparing SR and Comb for BCLC-B HCC, this study found no significant differences in RFS and OS between the groups investigated.

Systemic therapy with subsequent locoregional therapy has recently been proposed as a new therapeutic option for intermediate HCC [29,30]. The TACTICS trial showed that progression-free survival was prolonged by 9.7 months (median 24.9 months) and OS by 3.7 months (median 35.6 months) in HCC patients within the up-to-7 criteria who underwent sorafenib-TACE sequential therapy as compared with TACE alone [31]. Kudo reported an OS of 37.9 months for BCLC-B patients with upfront Lenvatinib following by selective TACE as compared with those treated only with TACE (21.3 months) (hazard ratio 0.48, 95% CI: 0.16–0.79, *p* < 0.01) [32]. In 2020, atezolizumab plus bevacizumab combination therapy (Atez/Bev) was approved following positive results obtained in the Phase 3 IMbrave150 trial [33]. Thereafter, Finn reported an overall response rate (RECIST ver. 1.1) of 44% in patients with BCLC-B HCC who were treated with Atez/Bev [34]. Thus, combination therapy including systemic treatment and another conventional therapeutic modality is now considered to be a promising strategy to improve the prognosis of BCLC-B HCC patients, although the rate for obtaining cancer-free status is considered to be inadequate at this time. The present results indicate that there are some BCLC-B cases that can be treated aggressively with SR including complementary use of RFA to achieve cancer-free status and prolong prognosis, although under limited conditions. A prospective trial should be conducted in the future to elucidate the clinical features of BCLC-B patients who are likely to achieve a better outcome by such aggressive therapeutic approaches.

This study has several limitations, including the fact that it is retrospective. As a result, there was selection bias regarding the treatment choice for HCC, as recent developments of SR (e.g., laparoscopic hepatectomy) were not taken into consideration. Additionally, number of patients in the Comb treatment (SR and RFA) group was low. Another limitation of this study is that results of the SR and Comb groups were not compared with those of patients who underwent TACE or systemic therapy, as proposed in the BCLC strategy. In the near future, a randomized control trial with a larger number of patients will be needed to obtain more concise conclusions regarding the usefulness of SR or Comb for BCLC-B HCC.

## 5. Conclusions

In the present study, RFS and OS were not significantly different between the SR and Comb groups. For patients in both groups with BCLC-B HCC beyond the up-to-7 criteria as well, there was no significant difference in RFS and OS, while the five-year survival rate for each was greater than 50%. Although the BCLC strategy recommends surgical resection for only BCLC-0/A HCC, the present results indicate that there are some patients with BCLC-B HCC for whom aggressive treatment with SR or Comb should be considered. Finally, it will be necessary to combine the advantages of SR and RFA with a flexible mindset rather than sticking to a single treatment modality.

## Figures and Tables

**Figure 1 cancers-15-00236-f001:**
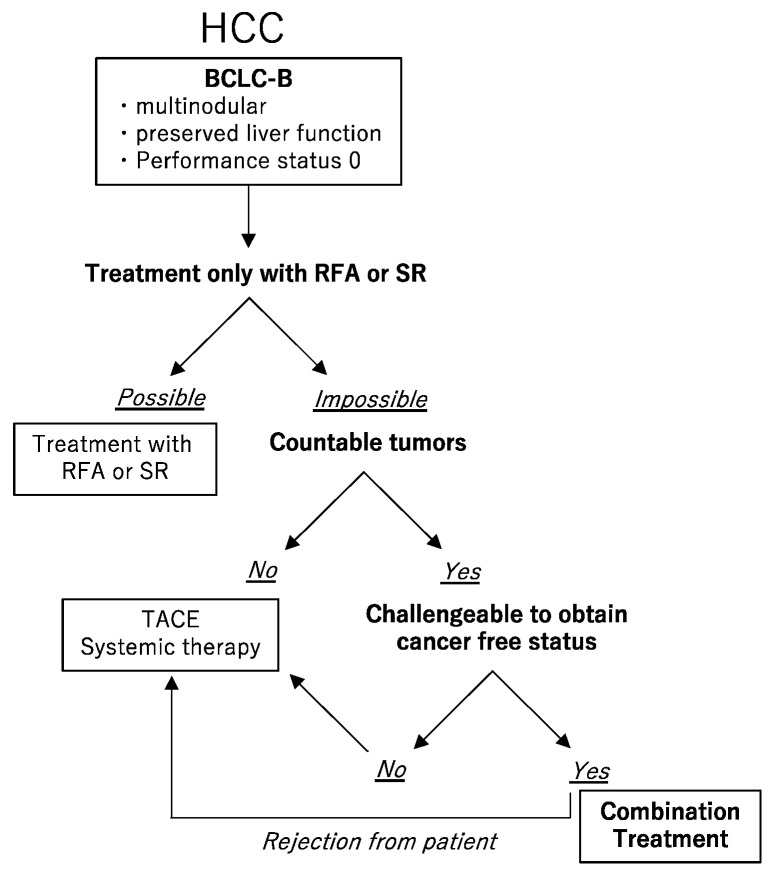
The choice of treatment. For the Barcelona Clinic Liver Cancer (BCLC) B hepatocellular carcinoma (HCC), if radiofrequency ablation (RFA) or surgical resection (SR) can be performed, they should be selected. If not, and if the number of HCCs cannot be counted, transarterial chemoembolization (TACE) or systemic therapy should be selected. If the number of HCCs can be counted, and if the cancer-free status can be challenged, in other words, if sufficient residual liver volume is available after removal of resectable tumors and the remained tumors are suitable for RFA, a combination of SR and RFA should be selected. Patients were properly informed of the treatment options including the guideline-based treatment, and if they did not wish to receive the combination treatment with SR and RFA, they received TACE or systemic therapy as per the guidelines.

**Figure 2 cancers-15-00236-f002:**
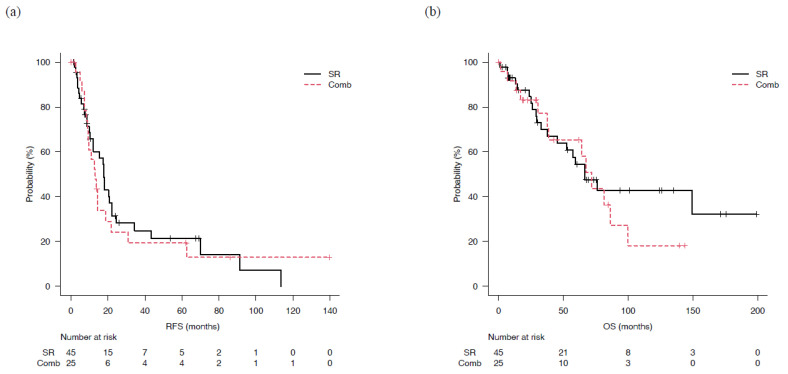
Recurrence-free and overall survival for patients who underwent surgical resection (SR) or combined surgical resection and radiofrequency ablation therapy (Comb). (**a**) There was not significantly difference of median recurrence-free survival (mRFS) in the SR (17.7 months, 95%CI: 10.3–22.1) and Comb groups (13.1 months, 95% CI: 9.0–18.9) (*p* = 0.70). (**b**) Median overall survival (mOS) of the SR and Comb groups was 66.6 (95% CI: 37.7-not applicable) and 72.0 (95% CI: 37.5–100) months, respectively, not significantly different (*p* = 0.544).

**Figure 3 cancers-15-00236-f003:**
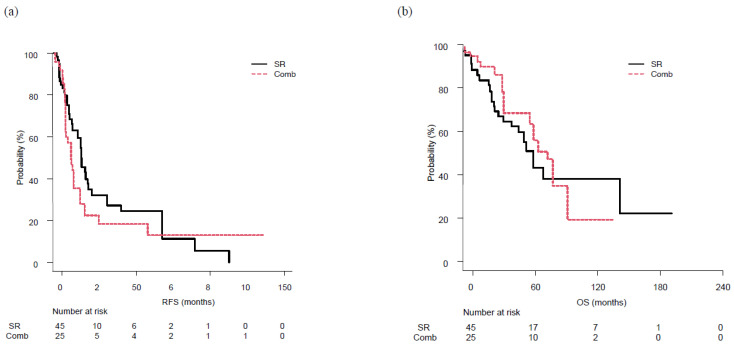
Recurrence-free survival (RFS) and overall survival (OS) following surgical resection (SR) or combined surgical resection and radiofrequency ablation therapy (Comb), following adjustment with inverse probability weighting. (**a**) RFS was similar in both the SR (17.7 months) and Comb (12.7 months) group, with no significant difference (*p* = 0.37). (**b**) Similarly, OS was similar in the SR (66.6 months) and Comb (67.5 months) group, with no significant differences (*p* = 0.65).

**Figure 4 cancers-15-00236-f004:**
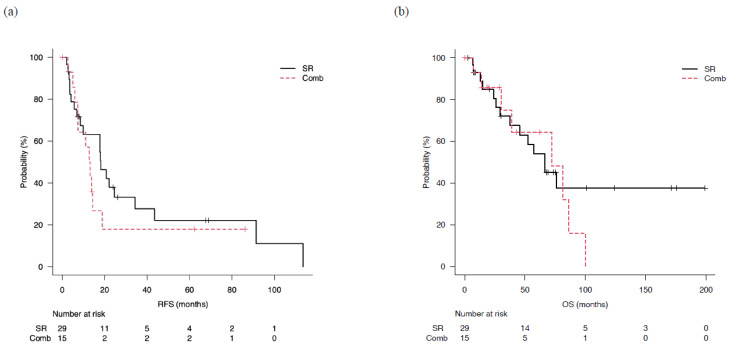
Outcomes in patients classified as beyond up-to-7 criteria who underwent surgical resection (SR) or combined surgical resection and radiofrequency ablation therapy (Comb). (**a**) Median recurrence-free survival (mRFS) in the SR and Comb groups was 18.2 (95% CI: 8.5–34.3) and 13.0 (95% CI: 5.9–18.9) months, respectively (*p* = 0.36). (**b**) Median overall survival (mOS) in the SR and Comb groups was 66.5 (95% CI: 29.4-not applicable) and 72.0 (95% CI: 30.2–86.2) months, respectively (*p* = 0.57).

**Table 1 cancers-15-00236-t001:** Patient characteristics (*n* = 70).

	All (*n* = 70)	SR Group (*n* = 45)	Comb Group (*n* = 25)	*p* Value
Age, years *	68 (63–74)	67 (64–75)	68 (62–73)	0.43
Gender, males:females	54:16	33:12	21:4	0.38
Etiology, HBV:HCV:HBV + HCV:alcohol:others	10:37:1:5:17	6:24:1::3:11	4:13:0:2:6	1.0
BMI, kg/m^2^ *	23.1 (28.1–25.1)	23.1 (22.1–25.2)	23.0 (21.7–24.0)	0.81
AST, U/L *	45 (30–77)	43 (29–60)	57 (30–86)	0.26
ALT, U/L *	42 (24–64)	41 (26–54)	50 (21–74)	0.43
Platelets, 10^4^/µL *	15.5 (11.3–19.1)	16.6 (12.9–19.3)	14.4 (9.7–16.4)	0.1
Total bilirubin, mg/dL *	0.7 (0.5–1.0)	0.7 (0.5–0.8)	0.9 (0.5–1.1)	0.1
Albumin, g/dL *	4.0 (3.7–4.2)	4.0 (3.8–4.3)	3.8 (3.6–4.2)	0.07
Prothrombin time, % *	88.4 (80.7–98.0)	90.7 (86.0–99.3)	84.6 (77.5–90.8)	0.01
ALBI score *	−2.66 (−2.95 to −2.44)	−2.74 (−3.02 to −2.57)	−2.52 (−2.90 to −2.26)	0.02
mALBI, 1:2a:2b:3	41:19:9:1	31:11:3:0	10:8:6:1	0.03
Child-Pugh score, A:B	67:3	45:0	22:3	0.04
FIB4-index *	3.4 (2.5–4.3)	3.3 (2.3–4.0)	4.2 (3.0–6.9)	0.03
AFP, ng/mL *	35.7 (6.9–340.2)	40.9 (7.9–841.1)	32.7 (4.8–68.0)	0.11
DCP, mAU/mL *	739 (150–3462)	673 (166–2737)	1210 (132–3884)	0.73
Tumor location (one:both lobes)	35:35	29:16	6:19	<0.01
Tumor size (maximum), cm *	4.95(4.0–6.0)	5.3 (4.0–7.0)	4.8 (3.7–5.8)	0.1
Number of tumors *	2 (2–3)	2 (2–3)	2 (2–3)	0.3

* Median. Values in parentheses show interquartile range, unless otherwise indicated. SR: surgical resection, Comb: combined SR and radiofrequency ablation, HBV: hepatitis B virus, HCV: hepatitis C virus, BMI: body mass index, AST: aspartate aminotransferase, ALT: alanine aminotransferase, ALBI score: albumin–bilirubin score, mALBI grade: modified ALBI grade, AFP: alpha-fetoprotein, DCP: des-gamma-carboxy prothrombin.

**Table 2 cancers-15-00236-t002:** Characteristics of patients classified as beyond up-to-7 criteria (*n* = 44).

	SR Group (*n* = 29)	Comb Group (*n* = 15)	*p* Value
Age, years *	66 (64–75)	64 (61–73)	0.36
Gender, males:females	23:6	13:2	0.70
Etiology, HBV:HCV:alcohol:others	3:13:3:10	2:7:2:4	1.0
BMI, kg/m^2^ *	23.1(22.1–25.0)	23.7 (21.2–26.7)	0.78
AST, U/L *	46 (36–62)	54 (30–78)	0.75
ALT, U/L *	42 (27–57)	40 (22–65)	0.94
Platelets, 10^4^/µL *	17.1 (14.9–21.5)	15.2 (14.0–17.7)	0.08
Total bilirubin, mg/dL *	0.8 (0.5–0.9)	0.7 (0.4–1.0)	1.0
Albumin, g/dL *	4.1 (3.8–4.4)	3.8 (3.6–4.2)	0.11
Prothrombin time, % *	90.0 (85.0–96.3)	79.0 (75.6–95.5)	0.12
ALBI score *	−2.74 (−3.07 to −2.57)	−2.52 (−2.95 to −2.26)	0.13
mALBI, 1:2a:2b:3	21:6:2:0	6:5:3:1	0.09
Child-Pugh score, A:B	29:0	12:3	0.03
FIB4-index *	3.1 (2.3–3.7)	4.0 (2.6–6.6)	0.13
AFP, ng/mL *	34.8 (6.8–977.2)	37.0 (8.4–88.2)	0.68
DCP, mAU/mL *	831 (169–13,293)	2545 (1246–17,802)	0.32
Tumor location (one:both lobes)	15:4:10	2:0:13	<0.01
Tumor size (maximum), cm *	6.3 (5.3–8.8)	5.5 (4.9–6.4)	0.08
Number of tumors *	2 (2–4)	3 (2.5–4)	0.34

* Median. Values in parentheses show interquartile range, unless otherwise indicated. SR: surgical resection, Comb: combined SR and radiofrequency ablation, HBV: hepatitis B virus, HCV: hepatitis C virus, BMI: body mass index, AST: aspartate aminotransferase, ALT: alanine aminotransferase, ALBI score: albumin–bilirubin score, mALBI grade: modified ALBI grade, AFP: alpha-fetoprotein, DCP: des-gamma-carboxy prothrombin.

## Data Availability

Due to the nature of this research, the participants could not be contacted regarding whether the findings could be shared publicly, thus supporting data, including datasets generated and/or analyzed for the current study, are not publicly available.

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
