# Peer review of "Clinical Usefulness of Surgical Resection Including the Complementary Use of Radiofrequency Ablation for Intermediate-Stage Hepatocellular Carcinoma"

_cancers, 2022, doi:10.3390/cancers15010236_

Round 1

Reviewer 1 Report

Ohama et al. proposed an interesting paper concerning treatment of HCC, notably in BCLC B classification.

Their results are rather good, comparing to classically recommended treatments in this situaton but disappointing in regard to first BCLC stages.

Nevertheless, some points must clarified:

- The way to the choice of treatment must be explained.

- BCLC classification and treatment choice are rather more a help to the treatment choice than a strictly recommendation. It would be interesting than the authors can discuss the BCLC limits.

Reviewer 2 Report

A study to look at combined liver resection and RPA for BCLC-B HCC.

Few queries:

1. The details of the RPA procedures were not mentioned apart from they were done percutaneously. Were they done before or after the liver resections?? Any of the cases were done in one setting at surgery. If the liver resection and RPA were done at two different settings, what is the time lag between the procedures and the imaging protocols to ensure the completeness of the ablation?! The potential long delay of the definitive treatment could generate significant selection bias in the study design.

2. More details of the tumours for RPA would be welcomed, in particular the size and location. In addition, what was the local recurrence rate?? What was the protocol for incomplete ablation?? It is important to know the success rate for the ablation procedure.

Reviewer 3 Report

We read with interest the paper entitled. “Clinical usefulness of surgical resection including complementary use of radiofrequency ablation for intermediate stage hepatocellular carcinoma” by Ohama et al. This study focuses on the outcome of surgery alone compared to surgery plus  radiofrequency ( RF) in patients with HCC belonging to stage BCLC -B.  The authors concluded that both approaches showed similar recurrence free survival (RFS) and overall survival (OS) with comparable side effects. In particular, the authors stressed the fact that both therapies offered a cumulative five-years OS ( > 50%) greater than offered by TACE ( the first-line treatment for HCC in BCLC-B stage)

This is a further contribution to a very long list of papers addressing the issue of the efficacy of surgery in BCLC patients with a well preserved liver function and supporting the so-called stage migration of therapeutic decisions in those patients.

Unfortunately the study is retrospective and lacks of granularity in selection of patients included.  It is well known that BCLC stage includes patients very different each other in terms of oncologic stage, liver function al clinical conditions. For these reasons, BCLC-B stage was sub-staged in 4 categories to better classify patients and chose proper treatments.  

The authors should therefore include information on the following aspects:

a- how many patients treated with surgery alone had single HCC, (size and location  should be detailed) and why they were classified as BCLC-B ( single HCC can be classified as BCLC-A regardless the size)

b- how many patients treated by the combo had single HCC ( size and location should be detailed), why the authors decided to perform ablation, and why those HCCs suitable for thermal ablation ( usually those < 3 cm) were classified as BCLC-B. In addition, how many RF sessions were needed  before surgery in each case

c- how many HCCs larger than 3 cm underwent RF and which was the final result achieved.

d- in the group of SR, the authors should detail how many patients had multifocal HCCs, where the lesions were located and which kind of resection was performed and why they decided not to perform RF prior surgery .

e- it should be clearly stated if RF was performed prior to  or during resection.

All these items should be included in the results and commented in Discussion.

Language and style are quite good and do not deserve large improvement.
